# Use of Vegetable Oils to Improve the Sun Protection Factor of Sunscreen Formulations

**Lucia Montenegro *** and **Ludovica Maria Santagati**

Department of Drug Sciences, University of Catania, Viale A. Doria 6, 95125 Catania, Italy;
ludovica.santagati@icloud.com

*   Correspondence: lmontene@unict.it; Tel.: +39-095-7384010

**Abstract:** Some vegetable oils have many biological properties, including UV-absorbing capacity. Therefore, their use has been suggested to reduce the content of organic UV-filters in sunscreen products. In this work, we investigated the feasibility of developing oil-based vehicles with a high sun protection factor (SPF) using pomegranate oil (PMG) and shea oil (BPO) in association with different percentages of organic UV-filters (octyl– methoxycinnamate, butyl methoxydibenzoylmethane, and bemotrizinol). We characterized the spreadability, occlusion factor, pH, and required hydrophilic lipophilic balance of the resulting formulations, and did not observe relevant differences due to the incorporation of vegetable oils. The in vitro spectrophotometric determinations of SPF values highlighted that the addition of BPO (1% (w/w)) and PMG (1% (w/w)) resulted in an increase in SPF in comparison with the same formulations that contained only organic UV-filters. The SPF increase was more significant for the formulations that contained lower amounts of organic UV-filters. The results of this study supported the hypothesis that including suitable vegetable oils in sunscreen formulations could be a promising strategy to design products with a lower content of organic UV-filters.

**Keywords:** sunscreens; vegetable oils; sun protection factor; oil-based vehicles; occlusion; spreadability

---

## 1. Introduction

The demand for cosmetic products based on natural ingredients is constantly growing, thus prompting researchers to find new compounds that could fulfil the requirements of consumers. Therefore, a significant amount of attention has been focused on vegetable oils, owing to their many biological activities and attractive technological properties, such as easy skin absorption and good spreadability [1–3]. Recently, selected herbal oils have been investigated to develop sunscreen formulations with a reduced content of organic UV-filters while still maintaining a high sun protection factor (SPF) [4–6]. Indeed, the safety of organic UV-filters has been widely debated because some studies on the skin penetration and permeation of these molecules have reported that they are able to permeate the skin at sufficient rates to be recovered from systemic circulation [7–9]. Therefore, to address the safety issue of organic UV-filters, several researchers performed in vivo toxicity studies from which the margin of safety (MoS) was calculated by comparing the potential human systemic exposure with the no adverse effect level (NOAEL) [10,11]. These studies revealed that the MoS of the investigated sunscreen agents was high enough to guarantee their safety after topical application in humans. These conclusions were also supported by in vitro studies pointing out that the penetration through human skin from the mineral oil of widely used sunscreen agents is so poor that it could not lead to any toxic effects [12].

The use of vegetable oils with photoprotective activity in sunscreen formulations would allow for decreasing the amounts of organic UV-filters, thus reducing the safety concerns and fulfilling the consumer demand for more natural products. Many herbal oils have been reported to show significant SPF values. Using an in vitro method, Kaur and Saraf [4] revealed that the SPF value for both olive oil and coconut oil was around 8, while that of castor oil and almond oil was around 6 and 5, respectively, thus highlighting the usefulness of these oils as UV-filters. Cosmetic O/W creams containing basil essential oil (5% (w/w)) have been proposed as potential sunscreen formulations for cosmetic and cosmeceutical uses [5]. Andréo-Filho et al. [6] investigated the photoprotective effect of solid lipid nanoparticles loaded with an organic sunscreen agent (octyl methoxycinnamate) and urucum oil, and concluded that the use of vegetable oils could allow sunscreen formulations to contain reduced amounts of organic UV-filters while maintaining the same SPF.

Other vegetable oils and plant or fruit extracts have been reported to show antioxidant and photoprotective effects [13–15]. Recently, pomegranate (*Punica granatum*) extracts have gained significant attention because of their beneficial effects, including anti-inflammatory, anticancer, and antimicrobial activity [16]. In addition, in vitro studies on cell cultures of normal human epidermal keratinocytes highlighted the protective effects of pomegranate fruit extract against UV-A and UV-B radiation [17,18]. The widespread cosmetic ingredient, shea butter (*Butyrospermum parkii* or *Vitellaria paradoxa*), has also been credited as being responsible for several effects after topical application, including photoprotective activity, but its real efficacy as a sunscreen agent has not yet been clearly demonstrated [19,20].

Therefore, in this work, we investigated the feasibility of using pomegranate and shea oil as natural UV-filters to improve the SPF value of sunscreen formulations.

As widely reported in the literature, vehicles can strongly affect the performance of cosmetic products, including sunscreen formulations [8,9,21–26]. The effectiveness of sunscreens and their compliance with consumer expectations requires that the product be easily spread onto skin, leaving a thin, uniform, adhering, and water-resistant film on the cutaneous surface. Among the different vehicles that could be used to prepare sunscreen formulations (O/W and W/O emulsions, oily lotions, and hydroalcoholic and oil-based vehicles), oil-based vehicles are most suitable for fulfilling the abovementioned requirements, owing to their ability to form a continuous and long-lasting film on the skin, and to their emollient properties that prevent skin dehydration caused by environmental exposure.

Hence, pomegranate and shea oil were incorporated into oil-based vehicles containing different percentages of organic UV-filters, and the technological properties as well as in vitro SPF values of the resulting cosmetic formulations were assessed.

## 2. Materials and Methods

### 2.1. Materials

Tocopheryl acetate (TA), argan oil, ethylhexyl stearate (EHS), C12–15 alkyl benzoate (Acemoll TN), wheat germ oil (WGO), sorbitan monooleate (Span 80), and polyoxyethylene sorbitan monooleate (Tween 80) were purchased from ACEF (Fiorenzuola d'Arda, PC, Italy). Span 85 was obtained from Sigma-Aldrich (Milan, Italy). Pomegranate seed oil (PMG) was supplied by Aroma-Zone (Cabrières-d'Avignon, France). Coconut alkanes (and) coco-caprylate/caprate (Vegelight 1214LC) was obtained from Grant Industries (Elmwood Park, NJ, USA).

BASF (Ludwigshafen, Germany) provided propylheptyl caprylate (Cetiol Sensoft), coco-caprylate (Cetiol C5), dicaprylyl ether (Cetiol OE), bis-ethylhexyloxyphenol methoxyphenyl triazine (Tinosorb S, BEMT), ethylhexyl methoxycinnamate (Uvinul MC 80, OMC), and butyl methoxydibenzoylmethane (Uvinul BMBM, BMBM). *Butyrospermum parkii* oil (BPO) was purchased from Esperis (Milan, Italy).

### 2.2. Oil-Based Vehicle Preparation

The compositions (% (w/w)) of the oil-based sunscreen vehicles investigated in this study are reported in Table 1. We chose organic UV-filter concentrations that would obtain vehicles with high SPF values and simultaneously guarantee UVA protection according to the data obtained using the BASF Sunscreen Simulator (www.basf.com/sunscreen-simulator). We adjusted the percentages of oil ingredients in the formulations containing higher percentages of UV-filters to obtain vehicles with similar technological properties. The oil-based vehicles were prepared by mixing all oily ingredients, including liquid UV-filters, at room temperature. Then, the solid organic UV-filters (BEMT and BMBM) were added and the mixture was stirred until a clear solution was obtained. The resulting formulations were stored at room temperature, in airtight glass containers, and sheltered from the light. The pH value of oil-based vehicles were determined 24 h after their preparation using a calibrated pH-meter (Crison pH-meter, model Basic 20, Crison Instruments, Alella, Barcelona, Spain) at room temperature.

**Table 1.** Composition (% (w/w)) of sunscreen oil-based vehicles. EHS: ethylhexyl stearate; TA: tocopheryl acetate; WGO: wheat germ oil; BPO: shea oil; PMG: pomegranate seed oil; BEMT: bis-ethylhexyloxyphenol methoxyphenyl triazine; BMBM: butyl methoxydibenzoylmethane; OMC: ethylhexyl methoxycinnamate.

| Ingredient | % (w/w) | | | | | | | | | | | |
|---|---|---|---|---|---|---|---|---|---|---|---|---|
| | Formulation Code | | | | | | | | | | | |
| | Ac1 | A1 | Ac2 | A2 | Bc1 | B1 | Bc2 | B2 | Cc1 | C1 | Cc2 | C2 |
| Acemoll TN | 17 | 16 | 16 | 15 | 16 | 16 | 16 | 14 | 17 | 16 | 16 | 15 |
| Cetiol C5 | 12 | 12 | 12 | 12 | 12 | 12 | 12 | 12 | 12 | 12 | 12 | 12 |
| Cetiol Sensoft | 10 | 10 | 10 | 10 | 10 | 10 | 10 | 10 | 10 | 10 | 10 | 10 |
| Vegelight | 20 | 19 | 19 | 18 | 20 | 18 | 18 | 18 | 18 | 17 | 18 | 17 |
| Cetiol OE | 14 | 14 | 14 | 14 | 14 | 14 | 14 | 14 | 14 | 14 | 13 | 13 |
| EHS | 6 | 6 | 6 | 6 | 6 | 6 | 6 | 6 | 6 | 6 | 6 | 6 |
| TA | 1 | 1 | 1 | 1 | 1 | 1 | 1 | 1 | 1 | 1 | 1 | 1 |
| Argan oil | 1 | 1 | 1 | 1 | 1 | 1 | 1 | 1 | 1 | 1 | 1 | 1 |
| WGO | 1 | 1 | 1 | 1 | 1 | 1 | 1 | 1 | 1 | 1 | 1 | 1 |
| BPO | - | 1 | - | 1 | - | 1 | - | 1 | - | 1 | - | 1 |
| PMG | - | 1 | - | 1 | - | 1 | - | 1 | - | 1 | - | 1 |
| BEMT | 5 | 5 | 5 | 5 | 6 | 6 | 6 | 6 | 7 | 7 | 7 | 7 |
| BMBM | 3 | 3 | 5 | 5 | 3 | 3 | 5 | 5 | 3 | 3 | 5 | 5 |
| OMC | 10 | 10 | 10 | 10 | 10 | 10 | 10 | 10 | 9 | 9 | 10 | 10 |

### 2.3. Determination of Required Hydrophilic Lipophlic Balance (rHLB)

To determine the rHLB values of the oil-based vehicles, we prepared a series of jars containing non-ionic surfactant blends with HLB values ranging from 2 to 14 (Span 80 8%/Span 85 92% HLB = 2; Span 80 88%/Span 85 12% HLB = 4; Span 80 83%/Tween 80 17% HLB 6; Span 80 65%/Tween 80 35% HLB 8; Span 80 46%/Tween 80 54% HLB10; Span 80 28%/Tween 80 72% HLB 12; Span 80 9%/Tween 80 91% HLB 14). Then, for the same oil-based vehicle, we prepared seven simple emulsions, each with the same amount of oil-based vehicle (20% (w/w)), the same amount of surfactant blend (5% (w/w)) but with a different HLB value, and the same amount of water (75% (w/w)). After mixing, we evaluated which emulsion appeared to be the most stable. The HLB of the surfactant blend used in the most stable emulsion was considered the rHLB value of that oil-based vehicle.

### 2.4. Spreadability

Sample spreadability was determined using the parallel plate method [27–29]. Two circular glass plates (diameter 9 cm) were used. One gram of the sample was placed on the lower plate and covered with the upper plate. Then, a 200 g weight was placed in the center of the upper plate, and after 1 min, the spreading diameter reached by the sample was measured (in centimeters). The spreadability was expressed as percentage (S%) according to the following equation:

$$S\% = (A2/A1) \times 100, \tag{1}$$

where A1 was the surface area of the glass plate and A2 was the surface area covered by the oily formulation after 1 min.

## 2.5. Occlusive Properties

In vitro occlusion tests were performed according to a previously reported method [30,31]. Beakers (100 mL) were filled with 50 mL of water, covered with filter paper (cellulose acetate filter, 90 mm, cutoff size: 4–7 μm, VWR, Fontenay sous Bois, France), and sealed. Then, 200 mg of the sample was uniformly spread onto the filter surface (18.8 cm$^2$; applied amount 10.6 mg/cm$^2$). The samples were then weighed and stored at 32 °C (50%–55% RH) for 48 hours. At the end of experiment, the samples were weighed to determine the water loss due to evaporation. Beakers covered with filter paper but without any applied sample were used as a reference. Each experiment was performed in triplicate. The occlusion factor (F) was calculated according to Equation (2):

$$F = 100 \times [(A - B)/A] \tag{2}$$

where A is the water loss without the sample (reference) and B is the water loss with the sample.

## 2.6. In Silico and In Vitro Sun Protection Factor (SPF) Determination

The sun protection factor (SPF) is the ratio of the minimum ultraviolet dose that produces erythema with and without sunscreen. According to Sayre et al. [32], this ratio can be conceptualized as the ratio of areas between the erythemal weighted solar radiation intensity with and without sunscreen. This basic concept has been applied to develop the BASF Sunscreen Simulator (www.basf.com/sunscreen-simulator) that we used to calculate the SPF values of oily formulations containing organic UV-filters.

In vitro SPF values of oily formulations containing vegetable oils and/or organic UV-filters were calculated spectrophotometrically according to Equation (3):

$$\text{SPF}_{\text{spectrophotometric}} = \text{CF} \times \sum_{290}^{320} \times \text{EE}(\lambda) \times \text{I}(\lambda) \times \text{Abs}(\lambda), \tag{3}$$

where CF is the correction factor (= 10), EE($\lambda$) is the erythemal effect of the radiation with wavelength $\lambda$, I($\lambda$) is the solar intensity of radiation with wavelength $\lambda$, and Abs($\lambda$) is the absorbance of the sunscreen product at wavelength $\lambda$.

Sayre et al. [32] determined the relationship between the erythemal effect EE($\lambda$) and the radiation intensity I($\lambda$) at each wavelength. These authors reported that the values of EE($\lambda$) × I($\lambda$) were constant at each wavelength (see Table 2). The absorption spectra of oily sunscreen formulations were obtained using a spectrophotometer (Shimadzu mod. UV-1601, Milan, Italy). The samples of each formulation were appropriately diluted with ethanol, and the same vehicle without organic and vegetable UV-filters was used as the blank (after dilution with ethanol as well as the related sample) to avoid interference due to the vehicle components. The absorption data were acquired every 5 nm in the range 290–320 nm, and three determinations were made for each sample.

**Table 2.** Relationship between erythemal effect (EE) and radiation intensity (I) at each wavelength ($\lambda$).

| $\lambda$ | EE × I |
|---|---|
| 290 | 0.0150 |
| 295 | 0.0817 |
| 300 | 0.2874 |
| 305 | 0.3278 |
| 310 | 0.1864 |
| 315 | 0.0839 |
| 320 | 0.0180 |
| Total | 1 |

## 3. Results and Discussion

Octyl methoxycinnamate (OMC) and butyl methoxydibenzoylmethane (BMBM) are among the most common UV-filters found in cosmetic sunscreen products. Although BMBM is one of the most commonly used UVA filters due to its strong absorption at 360 nm, this molecule shows poor stability after exposure to UV radiation [33,34]. When BMBM is in the presence of OMC, BMBM photodegradation products can react with OMC, thus reducing the photoprotective activity of OMC [35]. Chatelain and Gabard [36] demonstrated that the addition of bis-ethylhexyloxyphenol methoxyphenyl triazine (bemotrizinol, BEMT) to mixtures of BMBM and OMC was able to prevent BMBM photodegradation, thus improving the efficacy of sunscreen formulations containing BMBM and OMC. Therefore, in this work, we chose to use different percentages of BMBM and OMC in combination with BEMT to prepare oil-based vehicles with high SPF values.

It is well known that consumer acceptance is a key factor in determining the success of a cosmetic product [29,37]. As the sensory properties of the formulation strongly affect consumer opinion, the determination of parameters influencing both the efficacy and aesthetic characteristic of the final product plays a fundamental role in the design of a cosmetic formulation.

As far as sunscreen formulations are concerned, features such as spreadability, occlusion factor, and polarity may significantly influence both the efficacy and consumer acceptance of the product. Such parameters, as well as pH values, are reported in Table 3 for the oil-based vehicles Ac1, A1, Ac2, A2, Bc1, B1, Bc2, and B2. We did not perform the evaluation of the previously mentioned features on the formulations Cc1, C1, Cc2, and C2 because these formulations proved unstable, since a slight precipitate was observed 48 h after their preparation.

**Table 3.** Technological parameters, pH, required hydrophilic lipophilic balance (rHLB), spreadability (S%), and occlusion factor (F) of sunscreen oil-based vehicles.

| Formulation | pH | rHLB | S% | F |
|:---:|:---:|:---:|:---:|:---:|
| Ac1 | 6.08 | 10 | $35.63 \pm 2.45$ | $26.55 \pm 14.38$ |
| A1 | 6.90 | 10 | $40.91 \pm 5.02$ | $26.84 \pm 13.45$ |
| Ac2 | 6.80 | 12 | $36.17 \pm 2.84$ | $21.94 \pm 0.40$ |
| A2 | 6.27 | 12 | $34.16 \pm 0.10$ | $24.39 \pm 3.59$ |
| Bc1 | 6.84 | 10 | $42.58 \pm 2.66$ | $34.94 \pm 1.20$ |
| B1 | 6.86 | 10 | $26.96 \pm 1.14$ | $27.14 \pm 2.43$ |
| Bc2 | 6.50 | 12 | $31.76 \pm 2.93$ | $28.82 \pm 11.45$ |
| B2 | 6.10 | 12 | $39.17 \pm 2.56$ | $16.95 \pm 4.00$ |

As shown in Table 3, the spreadability values for the oil-based vehicles under investigation ranged from 27% to 42.5%. The incorporation of vegetable oils (BPO and PMG) into the formulation Bc1 significantly decreased the spreadability value while changes lower than 10% were observed after adding BPO and PMG to the other formulations. The range of pH values for all oily formulations was 6.1–6.9, thus pointing out that these formulations had pH values compatible with that of the skin surface.

The in vitro determination of the occlusion factor (F) provides an estimation of the ability of the formulation to prevent water loss from the skin [30]. All of the oil-based vehicles under investigation showed similar F values, apart from formulations Bc1 and B2, which had the highest and the lowest values, respectively.

Oil polarity is a factor involved in consumer acceptance of cosmetic products as polar oils often improve both skin feel and cosmetic aesthetics. As reported in the literature [38], the required hydrophilic–lipophilic balance (rHLB) can be used to estimate the polarity of natural and synthetic oils. By determining the rHLB values of the oily formulations under investigation, we observed that the incorporation of vegetable oils did not affect such parameters. It is interesting to note that the formulations with the lowest content of BMBM had lower rHLB values in comparison to those whose

BMBM content was higher. Based on pH, spreadability, rHLB, and occlusion factor values, we selected the formulations Ac2, A2, Bc2, and B2 to carry out the in vitro and silico determination of SPF.

Prior to determining the in vitro SPF values of oily sunscreen formulations, we assessed the UV-absorbing capacity of BPO and PMG in the range 290–320 nm. BPO and PMG showed different UV-absorbing profiles in the range 290–305 nm, while both oils were not able to absorb UV radiation with wavelengths higher than 305 nm (Table 4). Both oils had the highest absorbing ability at 290 nm, but PGM was about two-fold more effective that BPO at this wavelength. At higher λ values, the UV-absorbing capacity of both BPO and PMG decreased with a different trend, as PGO absorbance showed a 40% reduction from 295 to 300 nm, while PGO absorbance only decreased by 20%. Therefore, as BPO and PGM showed different absorbing profiles, we used a 1:1 mixture of these oils to improve the SPF values of oil-based vehicles containing organic UV-filters.

**Table 4.** Absorbance values at different wavelengths of shea oil (BPO) and pomegranate oil (PMG).

| λ | BPO | PMG |
|---|---|---|
| 290 | 1.486 | 3.198 |
| 295 | 1.001 | 2.161 |
| 300 | 0.620 | 1.837 |
| 305 | 0.166 | 0.403 |
| 310 | 0.000 | 0.000 |
| 315 | 0.000 | 0.000 |
| 320 | 0.000 | 0.000 |

In Figure 1, we depicted the UV-absorbing profile of the oily formulations Ac2, A2, Bc2, and B2 in the UV-B range (290–320 nm). As expected, the UV-absorbing ability of the formulations containing vegetable oils and organic UV-filters (A2 and B2) did not show any difference in comparison with the formulations containing only organic filters (Ac2 and Bc2) at wavelengths higher than 300 nm. In the range 290–305 nm, the UV-absorbing capacity of formulations containing BPO and PMG was greater than that of the corresponding vehicles without vegetable oils. UV absorbance data in the range 290–320 nm were used to calculate in vitro SPF values as previously reported by others [39,40].

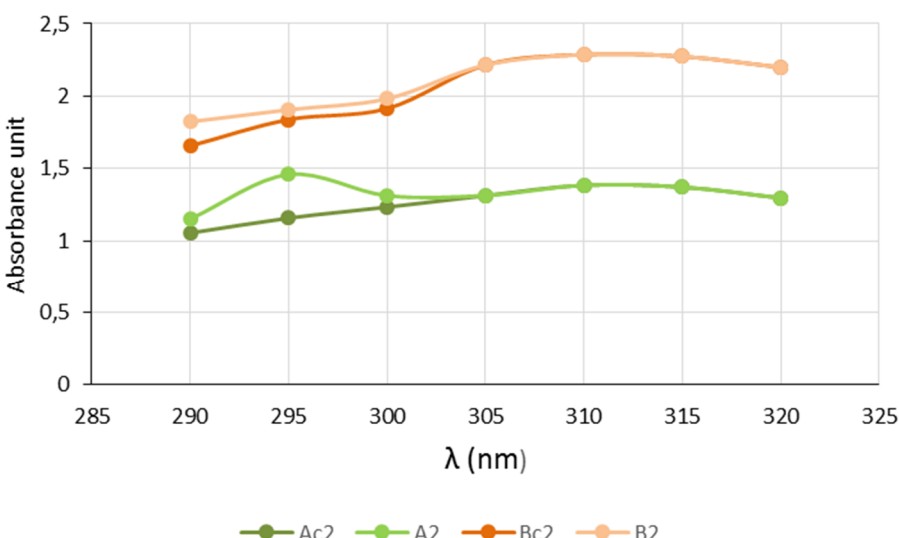

**Figure 1.** UV-absorbing profile of the oily formulations Ac2, A2, Bc2, and B2 in the range 290–320 nm.

As shown in Table 5, the in vitro SPF value of the formulation Bc2 was higher than that of Ac2 due to its higher BEMT content.

**Table 5.** In vitro sun protection factor (SPF) values of formulations containing only organic UV-filters (Ac2 and Bc2) or organic UV-filters and vegetable oils (A2 and B2) and the percentage of SPF enhancement (E%) of the formulation containing organic UV-filters and vegetable oils in comparison to the same formulation without vegetable oils.

| Formulation | SPF | E% |
| --- | --- | --- |
| Ac2 | 12.91 | — |
| A2 | 14.39 | 10.3 |
| Bc2 | 21.08 | — |
| B2 | 21.35 | 1.3 |

The addition of BPO and PMG to the formulation Ac2 provided a significant improvement in SPF value (10.3%), while only a very slight increase (1.3%) was observed when the same oils were included in the formulation Bc2. The lower photoprotective effect of BPO and PMG in the formulation containing a higher amount of organic UV-filters could be due to the interactions occurring between the vegetable oils and the synthetic sunscreen agents. Such interactions have already been reported for other vegetable oils in the literature [41].

SPF values calculated using the BASF Sunscreen Simulator for the oil-based vehicles Ac1, Ac2, Bc1, and Bc2 are reported in Figure 2. As only organic UV-filters were included in this SPF simulator, we could not determine the SPF values of the oil formulations containing vegetable oils using this in silico method. All oil-based vehicles showed high SPF values ranging from 34 to 39. The increase in BMBM percentage in the vehicle (formulation Ac1 vs. Ac2 and Bc1 vs. Bc2) did not significantly affect the resulting SPF value, while a higher percentage of BEMT (formulation Ac1 vs. Bc1 and Ac2 vs. Bc2) provided about a 10% improvement in SPF. It is interesting to note that for the same formulation, the SPF value determined spectrophotometrically and in vitro was significantly lower than that obtained in silico using the BASF Sunscreen Simulator. The discrepancy between the in vitro and in vivo predicted SPF values has been already pointed out by other authors who reported that interactions among sunscreen agents and vehicle components might interfere with the in vitro SPF spectrophotometric determination [42]. Therefore, the addition of vegetable oils to the oil formulations containing organic UV-filters could provide a higher increase in in vivo SPF values than was determined in vitro.

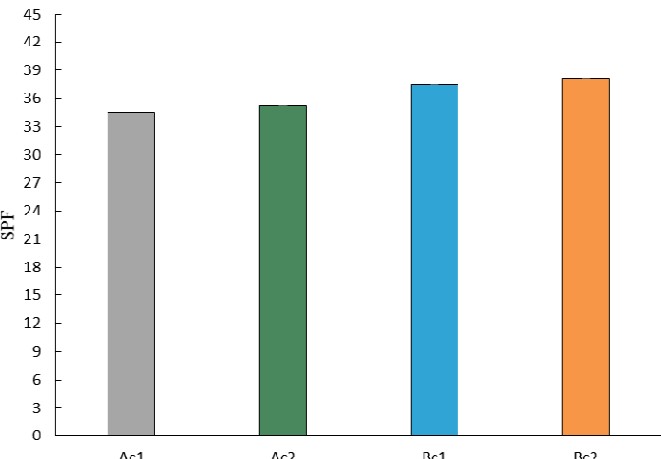

**Figure 2.** Sun protection factor (SPF) values of formulations Ac1, Ac2, Bc1, and Bc2 calculated using the BASF Sunscreen Simulator.

In conclusion, the results of our study highlight that the incorporation of pomegranate and shea oil into oil-based vehicles containing different percentages of organic UV-filters could be a useful strategy to improve SPF values. However, the ability of such vegetable oils to act as natural sunscreen

agents should be further investigated to assess their actual potential in different cosmetic vehicles, such as O/W emulsions.

In addition, Korkina et al. [41], reviewing the sun protection activity of secondary plant metabolites, suggested that the encapsulation of such plant derivatives into solid lipid nanoparticles (SLN) and nanostructured lipid carriers (NLC) could be a promising strategy to formulate sunscreen products with a lower organic UV-filter content. Several works have reported the successful loading of vegetable oils into SLN and NLC to improve the physicochemical and biological properties of these natural ingredients [43–46]. Therefore, further studies have been planned to assess the effect of co-loading pomegranate oil, shea oil, and organic UV-filters into lipid nanoparticles and the SPF of the resulting formulations.

**Author Contributions:** Conceptualization, L.M.; methodology, L.M.; validation, L.M., and L.M.S.; formal analysis, L.M.; investigation, L.M.S.; resources, L.M.; data curation, L.M.; writing—original draft preparation, L.M., and L.M.S.; writing—review and editing, L.M.; visualization, L.M.S.; supervision, L.M.; project administration, L.M.

**Funding:** This research received no external funding.

**Conflicts of Interest:** The authors declare no conflict of interest.

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
