# Peer review of "Use of Vegetable Oils to Improve the Sun Protection Factor of Sunscreen Formulations"

_cosmetics, doi:10.3390/cosmetics6020025_

Round 1

Reviewer 1 Report

Even though many - including researchers - use the terms chemical and physical filtesr, it remains wrong. All filters are chemicals! Originally, the terms chemical and physical filters have been "created" by marketiers when the ZnO and later TiO2 were introduced to discredit the organic filters already on the market!  Inorganic and organic filters would be better terms or best would be soluble or insoluble filters. Please change!

O/W and W/O oil emulsions (in the introduction) what is that. There are only O/W and W/O emulsions. And what is an oily lotion - that's an emulsion with a high proportion of lipophilic components. I think the authors should consult a up-to-date textbook on glacial formulations!

Table 1: What is the hydrophilic and what is the lipophilic phase? Please details in the table.

About pH: How did you measure pH - in the product or in the hydrophilic phase?

Author Response

Reviewer 1

Even though many - including researchers - use the terms chemical and physical filtesr, it remains wrong. All filters are chemicals! Originally, the terms chemical and physical filters have been "created" by marketiers when the ZnO and later TiO2 were introduced to discredit the organic filters already on the market!  Inorganic and organic filters would be better terms or best would be soluble or insoluble filters. Please change!

Answer

We thank the reviewer for reviewing our manuscript.

We agree with the reviewer that all filters are chemicals and inorganic and organic filters would be a better terminology. Therefore, we changed the term “chemical” into “organic” throughout the text.

O/W and W/O oil emulsions (in the introduction) what is that. There are only O/W and W/O emulsions. And what is an oily lotion - that's an emulsion with a high proportion of lipophilic components. I think the authors should consult a up-to-date textbook on glacial formulations!

Answer

We apologize for this typo. In the first draft of the manuscript, we explained the meaning of the abbreviations O/W and W/O but in the final draft we removed them because we thought they were redundant. Unfortunately, we missed to remove the last word of this explanation. We corrected the text accordingly.

We used the term oily lotions to indicate a formulation consisting only of oils, as shown in Table 1. However, due to the reviewer’s comment, we realized that the term we used is misleading. Therefore, we changed the term “oily lotions” into “oil-based vehicles” throughout the text.

Table 1: What is the hydrophilic and what is the lipophilic phase? Please details in the table.

Answer

As explained above, we used formulations consisting only of oil ingredients.

About pH: How did you measure pH - in the product or in the hydrophilic phase?

Answer

We measured pH values of the oil vehicles. As explained above, we did not have any aqueous phase.

Reviewer 2 Report

While authors performed an interesting study about the potential advantages of certain vegetable oils as organic SPF boosters, there are a few points of revision to be addressed.

Most of all, the last part of study using in silico SPF prediction method does not provide any meaningful information related with the research subject, as authors stated. The possible effects of oily ingredients (vegetable oils used in this study) could not be investigated by in silico analysis due to the basic conditions of program. So, this part of experiment does not give any information to readers, and it would be better to remove this part or simply state in one or two sentence. 

For the formulation information provided in Table 1, it should be stated why authors choose to modify each ingredients' concentration as such. 

The change of SPF value by introduction of vegetable oils, presented by Figure 2, is not easily acknowledged, possibly due to the small changes in SPF value. It would be better to present it as table with % increase data. 

There are a few typos needs to be revised, i.e. abbreviation of shea oil in abstract line 12 should be BPO, not PBO. 

Regards,

Author Response

Reviewer 2

While authors performed an interesting study about the potential advantages of certain vegetable oils as organic SPF boosters, there are a few points of revision to be addressed.

Most of all, the last part of study using in silico SPF prediction method does not provide any meaningful information related with the research subject, as authors stated. The possible effects of oily ingredients (vegetable oils used in this study) could not be investigated by in silico analysis due to the basic conditions of program. So, this part of experiment does not give any information to readers, and it would be better to remove this part or simply state in one or two sentence.

Answer

We thank the reviewer for reviewing our manuscript.

As the reviewer noticed, we pointed out that the effects of vegetable oils on SPF could not be determined using in silico SPF prediction method. However, we thought it was noteworthy to highlight that our in vitro SPF values were significantly lower than SPF values obtained using in silico analysis and expected in vivo. Indeed, our results support the observation of other authors who found that in vitro SPF values were lower than expected in vivo SPF values.  Hence, we consider it essential to point out this discrepancy between in vitro SPF and in vivo expected SPF to convey the hypothesis that the use of vegetable oils could provide higher SPF values than those obtained in vitro. To better clarify this concept, we added in the text the following sentence: “Therefore, the addition of vegetable oils to oil formulations containing organic UV-filters could provide an increase of in vivo SPF values higher than that determined in vitro.”

For the formulation information provided in Table 1, it should be stated why authors choose to modify each ingredients' concentration as such.

Answer

Thanks to the reviewer’s comment, we realized that we missed to mention why we chose the concentrations of UV-filters and oil ingredients reported in Table 1. Therefore, we added in the text (line 117) the following sentences: “We chose organic UV-filter concentrations to obtain vehicles with high SPF values and to simultaneously guarantee UVA protection, according to the data obtained using BASF sunscreen simulator (www.basf.com/sunscreen-simulator). We adjusted the percentages of oil ingredients in formulations containing higher percentages of UV-filters to obtain vehicles with similar technological properties. “

The change of SPF value by introduction of vegetable oils, presented by Figure 2, is not easily acknowledged, possibly due to the small changes in SPF value. It would be better to present it as table with % increase data.

Answer

As generally figures have a better visual impact than tables, we reported in vitro SPF data in a graph and we indicated in the text the percentage of SPF increase due to the presence of vegetable oils in the formulation.  However, to comply with the reviewer’s request, we reported in vitro SPF data and the percentage of SPF increase due to the presence of vegetable oils in the formulation in a Table. Therefore, we inserted Table 5 in the text (see below) and we modified the text accordingly.

Table 5. In vitro sun protection factor (SPF) values of formulations containing only organic UV-filters (Ac2 and Bc2) or organic UV-filters and vegetable oils (A2 and B2) and percentage of SPF increase of the formulation containing organic UV-filters and vegetable oils in comparison to the same formulation without vegetable oils.

Formulation

SPF

EF %

Ac2

12,91

---

A2

14,39

10,3

Bc2

21,08

---

B2

21,35

1,3

There are a few typos needs to be revised, i.e. abbreviation of shea oil in abstract line 12 should be BPO, not PBO.

Answer

We apologize for this typo and we have corrected it.

Reviewer 3 Report

This manuscript is devoted to implementation of vegetable oils into sunscreen formulations in order to reduce amounts of chemical UV-filters as octyl methoxycinnamate, butyl methoxydibenzoylmethane and bemotrizinol.

The results obtained in this work are promising, but if these oils are to be used for the anti-radiation filters, further studies on the encapsulation of natural oils will be necessary to perform.

Author Response

Reviewer 3

This manuscript is devoted to implementation of vegetable oils into sunscreen formulations in order to reduce amounts of chemical UV-filters as octyl methoxycinnamate, butyl methoxydibenzoylmethane and bemotrizinol.

The results obtained in this work are promising, but if these oils are to be used for the anti-radiation filters, further studies on the encapsulation of natural oils will be necessary to perform.

Answer

We thank the reviewer for reviewing our manuscript. We agree with the reviewer that further studies on the encapsulation of vegetable oils would be required to explore their actual potential as UV-filters. Therefore, in the conclusions (line 352), we had already highlighted that “further studies have been planned to assess the effect of co-loading pomegranate oil, shea oil and organic UV-filters into lipid nanoparticles on the SPF of the resulting formulations.”

Round 2

Reviewer 1 Report

sticking a pH probe into an o/w or w/o emulsion will certainly show a number on a display, however scientifically sound pH measurements are only possible in aqueous solutions! I guess you just delivered a "house number"!

Reviewer 2 Report

Nothing further to be revised.